# *Mucochytrium quahogii* (=QPX) Is a Commensal, Opportunistic Pathogen of the Hard Clam (*Mercenaria mercenaria*): Evidence and Implications for QPX Disease Management

**DOI:** 10.3390/jof8111128

**Published:** 2022-10-26

**Authors:** Sabrina Geraci-Yee, Jackie L. Collier, Bassem Allam

**Affiliations:** School of Marine and Atmospheric Sciences, Stony Brook University, Stony Brook, NY 11794, USA

**Keywords:** quahog parasite unknown, opportunistic pathogen, *Mercenaria mercenaria*, QPX disease, commensal, disease management, quantitative PCR, labyrinthulomycetes

## Abstract

*Mucochytrium quahogii*, commonly known as QPX (Quahog Parasite Unknown), is the causative agent of QPX disease in hard clams (*Mercenaria mercenaria*), but poor understanding of the relationship between host and pathogen has hindered effective management. To address this gap in knowledge, we conducted a two-year study quantifying the distribution and abundance of *M. quahogii* in hard clam tissue, pallial fluid, and the environment. *M. quahogii* was broadly distributed in clams and the environment, in areas with and without a known history of QPX disease. *M. quahogii* in clams was not strongly related to *M. quahogii* in the environment. *M. quahogii* was always present in either the tissue or pallial fluid of each clam, with an inverse relationship between the abundance in the two anatomical locations. This study suggests that the sediment–water interface and clam pallial fluid are environmental reservoirs of *M. quahogii* and that there is a host-specific relationship between *M. quahogii* and the hard clam, supporting its classification as a commensal, opportunistic pathogen. There appears to be minimal risk of spreading QPX disease to naïve clam populations because *M. quahogii* is already present and does not appear to be causing disease in hard clam populations in locations unfavorable for pathogenesis.

## 1. Introduction

*Mucochytrium quahogii*, commonly known as Quahog Parasite Unknown (QPX), is the causative agent of QPX disease in hard clams, *Mercenaria mercenaria* [1]. *M. quahogii* belongs to a diverse and ubiquitous group of marine protistan decomposers known as labyrinthulomycetes that are generally nonpathogenic but some of which have been reported as opportunistic pathogens of diverse marine animals [2,3,4]. With infectious marine diseases already on the rise [5], opportunistic infections are expected to increase with environmental change, as marine species may become physiologically challenged, stressed, and immunocompromised [2,6]. Marine diseases have the potential to cause ecosystem-wide impacts driven by mass mortality of ecologically and economically important species, such as the hard clam. Therefore, studying opportunistic pathogens and understanding their dynamics outside of their host are very important, as it may help clarify the cause of changes in the host–microbe relationship and the development of pathogenesis [2]. Knowledge on pathogen ecology will help to understand disease dynamics, including source(s), transmission, virulence, and environmental factors that initiate disease development. This information is needed to develop appropriate management methods, which are currently limited, as most epidemiological theory has been developed for terrestrial systems and does not account for the openness and spatial scale of the marine system [7]. Current disease-management practices for infectious diseases in humans and terrestrial wildlife are ineffective or prohibitively expensive in marine ecosystems, leaving most remediation efforts limited to reducing pathogen input [5]. This represents the current management practices for QPX disease, which may not be possible for opportunistic pathogens that can survive and replicate outside the host, and may be ubiquitous in the environment. Developing adapted control measures is difficult, given our lack of knowledge on most marine pathogens such as *M. quahogii*.

Research focused on its pathobiology has revealed that QPX disease is likely facilitated by specific environmental factors, such as low temperature and high salinity, that give *M. quahogii* an advantage in the host–pathogen interaction [2,3,8]. However, there remain major gaps in our knowledge about the basic biology and ecology of *M. quahogii.* The mechanism of QPX disease transmission is still unknown, as well as the distribution and abundance of *M. quahogii* in the environment. Based on histopathology results that show typical QPX disease infection primarily occurs in the pallial organs, the pallial fluid (i.e., the fluid within the mantle cavity of bivalves that surrounds the pallial organs) may represent the site of initial interactions between hard clams and *M. quahogii*, as well as other labyrinthulomycetes, from environmental reservoirs [9,10,11]. 

To date, there is only one survey that investigated the distribution of *M. quahogii* in hard clams and the environment, which used a very sensitive but non-quantitative PCR assay evaluated by denaturing gradient gel electrophoresis (DGGE) [12,13]. Seawater, sediment, seaweeds, seagrass, and various invertebrates, including hard clams, collected in Massachusetts and Virginia, each at some point during the two-year study tested positive for *M. quahogii*. There was a hint of seasonal pattern with *M. quahogii* most prevalent in seawater, seaweeds, and seagrass during the spring, seawater and sediment during the summer, and invertebrates (including hard clams) in the fall. There was little difference in detection of *M. quahogii* between sites with and without active QPX disease [13].

Thus, evidence to date suggests that *M. quahogii* is an opportunistic pathogen of the hard clam because it has been found in healthy and diseased hosts, and widely detected in the natural environment, with pathogenesis only occurring under environmental conditions detrimental to hard clams [14]. Yet, management of QPX disease in both cultured and wild hard clams has been focused on limiting its spread based on the assumption that *M. quahogii* is an obligate pathogen. Critically, there is not enough evidence to determine if the current QPX disease-management strategy is effective in preventing the spread of QPX disease among hard clam populations. If *M. quahogii* is a widespread opportunistic pathogen, the absence of QPX disease epizootics (mortality events) in locations with hard clam populations may be the result of environmental conditions unfavorable to disease development, rather than the absence of *M. quahogii* in those environments. If the presence of *M. quahogii* is not dependent on hard clams, insight into what environmental factors determine its distribution and abundance relative to other labyrinthulomycetes may support better management of QPX disease. 

To fill these gaps in knowledge on the distribution of *M. quahogii* in the hard clam habitat, an extensive two-year survey of *M. quahogii* in hard clams and the environment was conducted throughout the marine district of New York (NY), including areas with and without previous history of QPX disease in hard clams. This is the first field survey to quantitatively examine *M. quahogii* in hard clams, including both tissue and pallial fluid (with associated pseudofeces), and in the environment (seawater and sediment). Results showed that *M. quahogii* is broadly distributed in both host and the environment, in areas with and without a known history of QPX disease, suggesting that *M. quahogii* is a commensal, opportunistic pathogen of the hard clam. These findings have important implications for the management of QPX disease as there appears to be minimal risk of spreading QPX disease because *M. quahogii* is already present without causing disease in hard clam populations.

## 2. Methods

### 2.1. Sampling Sites

Hard clam and environmental samples were collected from Raritan Bay, Babylon Bay, Moriches Bay, Shinnecock Bay, Oyster Bay, Port Jefferson Harbor, Birch Creek, and Peconic Estuary (Figure 1) on a monthly rotating schedule (Appendix A) for a total of 72 sampling events during 2014 and 2015. Most times, both clams and environmental samples (sediment and seawater) were taken, but occasionally only clams or only environmental samples were collected. QPX disease was previously found in Raritan Bay, Oyster Bay, and Birch Creek, while the other sites had not previously been examined for QPX disease. Samples were also collected from a heavily QPX-impacted clamming site in Barnstable, Massachusetts (MA) (Figure 1), where QPX disease is considered enzootic. Since the initial QPX disease outbreak in 2002, Raritan Bay has been monitored, which has revealed a complex disease history [15]; therefore, we sampled 4 sites within the bay (20B and 21 in 2014; 8 and 16 in 2015). The central Raritan Bay sites 8, 16, and 21 are locations with high clam density (usually > 70 clams/m^2^), which have had a continuous presence of QPX disease over the last 20 years. In contrast, at site 20B in Great Kills Harbor, QPX disease is extremely low despite an even greater clam density (>250 clams/m^2^). We also included a site in the Peconic Estuary in our survey since it was historically used as the receiving bay for the New York Shellfish Transplant Fishery [16].

### 2.2. Sample Collection and Processing

At each sampling event, approximately 30 hard clams were collected using a bull or “bubble” rake. Clams were insulated by bubble-wrap and put on ice. Surface and bottom seawater were collected using a 2 L Niskin bottle (General Oceanics, Miami, FL, USA). Sediment was collected using a ponar sediment grab (0.04 m^2^) and surface sediment (top 2 cm) was homogenized and stored in a sterile Whirl-Pak^®^. All samples were put on ice for transport back to the laboratory for processing. Temperature, salinity, and dissolved oxygen of both surface and bottom seawater were measured with a Model 85 YSI sonde (YSI Inc., Yellow Springs, OH, USA). Water depth was also recorded, as well as the difference between surface and bottom seawater for the measured parameters. 

At the lab, hard clams were stored at 4 °C and processed within 24 h. Clams were gently washed and pallial fluid was collected first, if being sampled. Hard clam pallial fluid was only collected from samples during May to July in 2014 (*n* = 291). The clam was carefully opened to collect pallial fluid by placing a shucking knife in between the valves, avoiding cutting the muscle to prevent hemocytes from getting into the sample. The clam was held upright to drain and approximately 1 to 2 mL of pallial fluid was collected from each clam and stored at −80 °C until DNA extraction. When present, pseudofeces were also collected within the sample and documented. Clams were then shucked and examined for gross signs of QPX disease (e.g., was removed until approximately 250 µL of the supernatant and pellet remained to be used for DNA extraction. Pallial fluid samples were extracted following the same inflammatory nodules). Clams were then processed for qPCR and histology [17]. Briefly, clams were dissected and a thin cross-section of clam meat, containing mantle, siphon, gills, digestive glands, stomach, gonad, pericardium and kidney, was fixed in 10% buffered formalin to be used for histological examination [17,18]. Remaining mantle and siphon tissues were homogenized in 10 volumes (*w*/*v*) of 1X phosphate buffered saline (PBS) and stored at −80 °C until DNA extraction. A 200 µL aliquot (equivalent to 20 mg of tissue) was used for DNA extraction (after removal of PBS by centrifugation) using the NucleoSpin Genomic DNA Tissue kit (Macherey-Nagel, Inc., Bethlehem, PA, USA), following the standard protocol for animal tissue and eluted in 150 µL (2 elutions of the spin column using 75 µL of elution buffer). Pallial fluid samples were thawed, vortexed, and spun down at 12,000× *g* for 10 min. The supernatant protocol as the tissue samples, except the sample was eluted in only 100 µL. For both hard clam sample types, DNA quantity and quality were evaluated using Nanodrop ND-1000 spectrophotometry (Thermo Scientific, Wilmington, DE, USA) and DNA stored at −20 °C until assayed. 

Both tissue and pallial fluid (when sampled) from the same individual hard clam were assayed using a *M. quahogii*-specific qPCR in triplicate reactions using 1 µL of template DNA [17]. For each sampling time point, we randomly selected up to 22 clams from the cohort of up to 30 clams collected; at some sites, clams were scarce and therefore number of clams assayed from each cohort varied from 7 to 22 individual clams (mean ± SD = 16 ± 2) for a total of 977 clams in 59 cohorts. Since pallial fluid was a new sample type, PCR inhibition testing was performed by spiking in linear plasmid to create a dilution series (1:10) to measure PCR efficiency and linearity [16,19]. A representative group of samples was assessed, revealing that inhibition was minimal with PCR efficiency and linearity within acceptable limits (10% compared to the standard curve) [20]. *M. quahogii* (QPX) prevalence and weighted prevalence, according to the intensity scales in Appendix A, as well as concentration minima, maxima, mean, and range were determined for each clam cohort and sample type. *M. quahogii* prevalence (percent QPX-positive, denoted as TPOS = total positive) included positive quantifiable samples (POS) and samples that were positive but below the limit of detection, denoted as BLD [17]. Briefly, BLD samples were positive because they had a high threshold cycle value (Cq or Ct) product with the expected melting temperature, and a band of the expected size when evaluated by agarose gel electrophoresis. However, these samples were below the limit of detection of the qPCR (10 copies) so the target could not be quantified. Therefore, for these samples the qPCR assay functioned as a conventional PCR giving us only presence or absence. Based on qPCR results for tissue samples, hard clams with the highest *M. quahogii* signal were analyzed by histopathology to confirm active QPX disease.

Environmental samples (*n* = 206) were processed immediately upon return to the lab, as described by Geraci-Yee et al. (in review). Surface sediment (top 2 cm) was homogenized and stored at −80 °C until DNA extraction. Up to 350 mL of surface and bottom seawater samples (SSW and BSW, respectively) were filtered under low vacuum pressure (<5 in Hg) on a 0.4 µm (47 mm) polycarbonate filter (GE Osmonics Inc., Minnetonka, MN, USA) and stored at −80 °C until DNA extraction. Environmental DNA was extracted with the MO BIO PowerSoil DNA isolation kit (Qiagen, Hilden, Germany) according to the manufacturer’s protocol, eluted in 100 µL and stored at −20 °C until assayed. Samples were assayed in triplicate using the newly developed nested, quantitative PCR (nqPCR) assay for increased sensitivity, using 3 µL of template DNA [16,19]. Environmental samples were also assayed for total labyrinthulomycetes using a newly developed qPCR in triplicate using 1 µL of template DNA [16]. All quantitative data determined by qPCR (i.e., quantifiable results but not BLD) and nqPCR assays are expressed in terms of gene copies per mg tissue or sediment or mL pallial fluid or seawater. Conversion to *M. quahogii* cellular abundance is presented using 440 copies per mononucleate cell [17]. 

### 2.3. Labyrinthulomycete Diversity in Hard Clam Pallial Fluid

Despite inhibition being minimal in the *M. quahogii*-specific qPCR assay, there was strong inhibition of the general labyrinthulomycete PCR by hard clam pallial fluid samples. Therefore, an additional set of pallial fluid samples (*n* = 23) was extracted using the MO BIO PowerSoil DNA isolation kit (Qiagen, Hilden, Germany), which has additional steps for inhibitor removal. The 23 pallial fluid samples consisted of 2 to 5 samples from RB21, OB, BB, PE, SB, and MA collected during June or July 2014. DNA quantity was assessed with Quant-iT PicoGreen (Molecular Probes, Eugene, OR, USA) and a subset of the samples (*n* = 10) were evaluated by Nanodrop for DNA purity, which was better compared to the NucleoSpin extraction kit (average A260/280 ratio of 1.92 ± 0.35 SD compared to 1.66 ± 0.25). We used traditional end-point PCR and labyrinthulomycete-specific primers: LABY-A and LABY-Y [21]. Each 20 µL PCR reaction contained 2 µL of 25 mM MgCl_2_, 2 µL of 2 mM dNTPs, 2 µL of 2 µM forward and reverse primers, 4 µL of 5X GoTaq Flexi buffer (colorless), 0.5 µL GoTaq G2 Flexi DNA Polymerase (Promega, Madison, WI, USA), 1 to 6 µL of template DNA, with the remaining volume of nuclease-free water. Initial PCR trials used 1 µL of template DNA; if the PCR did not result in a product, the volume of template DNA was increased to 2 µL and then 6 µL to attempt to amplify a labyrinthulomycete PCR product. The labyrinthulomycete-specific PCR program used was 35 cycles of 95 °C for 30 s, 55 °C for 1 min, 72 °C for 1 min. PCR products were examined by agarose gel electrophoresis and the expected band was excised and cleaned using the Wizard SV Gel and PCR Clean-Up System (Promega, Madison, WI, USA). 

Gel-purified LABY-AY PCR products (~430 bp, single amplicons) were directly Sanger sequenced using the forward primer (LABY-A) on an ABI3130XL sequencer. The resulting chromatograms were visually examined and samples that were “clean”—containing single peaks—were directly used for downstream analyses. Samples that contained a mix of sequences (multiple peaks underneath the dominant peaks) were deconvoluted using ‘Base-Calling Algorithm with Vocabulary (BCV)’ [22], with a ‘dictionary’ of sequences containing known LABY-AY PCR products. All output sequences, referred to as clusters, with a 10% or greater expected contribution to the mixed chromatogram and at least half the length of the input sequence—usually 1 to 3 clusters per chromatogram—were used in downstream analyses. These sequences were compared against GenBank, using the default settings with the exclusion of uncultured/environmental sequences, with Basic Local Alignment Search Tool for nucleotide sequences (Blastn). To further validate the taxonomic identification of the sample by Blastn, all of the resulting top Blastn hits (non-redundant) were compiled with the pallial fluid sequences, and the resulting 90 sequences (25 from the pallial fluid samples and 65 reference sequences) were aligned with ClustalW in BioEdit v7.2.5 [23,24]. The alignment was then used to construct a Maximum Likelihood phylogenetic tree in MEGA X [25,26] based on 391 positions using a general time-reversible model, gamma distributed (5 categories) with invariant sites, nearest-neighbor interchange with 100 bootstrap replications. Based on the Blastn result and phylogenetic tree, each original pallial fluid sequence or BCV cluster was assigned a labyrinthulomycete phylogenetic group (i.e., labyrinthulid, aplanochytrid, oblongichytrid, thraustochytrid, or *M. quahogii*/QPX). A value of 1 was assigned for samples that had clean chromatograms, while for the BCV clusters the expected portion of the mixed sample, generated by BCV was used. 

### 2.4. Metadata

Chlorophyll content of SSW and BSW samples was determined using acetone extraction and spectrophotometry [27]. Up to 200 mL of seawater was filtered on a 25 mm GF/F filter (GE Whatman, Maidstone, UK) and the filter was folded and stored in aluminum foil at −80 °C until acetone extraction. Filters were placed in 5 mL of 90% acetone, vortexed, and stored in the dark at −20 °C for 24 h. Samples were centrifuged for 15 min at 500× *g* at 4 °C and supernatant was transferred to a 2 mL microtube and spun again at 16,000× *g* for 5 min to remove any remaining filter particles. The absorbance at 750, 664, 647, 630, and 600 nm was recorded, and chlorophyll a, b, and c were determined using equations for mixed phytoplankton assemblages [28].

Measured metadata were supplemented with local weather data from the closest weather station to the sampling site (Appendix A), including air temperature (minima, mean, and maxima), wind speed (maxima, mean, and gust), wind direction, cloud cover, precipitation, and weather event. In addition, for temperature, wind speed, and precipitation, the mean from 1 to 4 months prior to the sampling date was determined. For example, the mean 1-month temperature was calculated from the day of sampling to 30 days prior; the mean 2-month up to 60 days prior; the mean 3-month up to 90 days prior; and the mean 4-month up to 120 days prior. For precipitation, we also determined the sum of 1 to 4 months prior to the sampling date. Furthermore, we also determined the lag mean 2 to 4 month temperature. For the 2-month lag mean temperature, the average temperature was determined for 30–60 days prior to the sampling date; the 3-month lag mean was 60–90 days; and the 4-month lag mean was 90–120 days. 

### 2.5. Data Analyses

Data were visualized using Microsoft Excel or ggplot2 [29] in RStudio v1.2.1335 [30]. All quantitative data were evaluated for normality using Shapiro–Wilk’s normality test in R (v3.6.0), which revealed that data were distributed significantly different from normal; therefore, all statistical tests performed were non-parametric. Comparison of *M. quahogii* prevalence and abundance from clams and environmental samples under various groupings (e.g., QPX disease history, sampling year, site, or month) were evaluated by Wilcoxon rank sum test for 2 groups and Kruskal–Wallis rank sum test for more than 2 groups with Bonferroni (BF) correction for each set of comparisons. However, being an exploratory analysis, we considered potentially significant differences from both corrected and uncorrected *p*-values. Significant groupings by Kruskal–Wallis rank sum test were further investigated by pairwise comparisons using Wilcoxon rank sum test with and without Benjamini–Hochberg (BH/FDR) correction. Correlational analyses between prevalence or abundance data and metadata were performed using Spearman’s rank-order method. Again, as an exploratory analysis, we considered all correlations with *p* < 0.05 with or without BF correction. Correlations are described as weak (|rho| < 0.3), moderate (0.3 < |rho| < 0.5), and strong (|rho| > 0.5). Correlograms with histograms for visualization of correlations were created in RStudio using the Performance Analytics package [31]. Multivariate analyses were also performed on the data [16], but did not offer further insight.

## 3. Results

During the two-year field survey, there were 72 sampling time points with a total of 1183 samples comprising 977 clams in 59 cohorts (hard clam tissue samples) and 206 environmental samples (Table 1), including 71 sediment (SED), 64 bottom seawater (BSW), and 71 surface seawater (SSW) samples. *M. quahogii* (QPX) was detected in the majority of samples of all sample types (Figure 2) with no difference in prevalence (%QPX-positive samples) among sample types by Kruskal–Wallis rank sum test. 

### 3.1. Environmental Parameters

The measured environmental parameters: seawater temperature, salinity, dissolved oxygen (DO), and chlorophyll are presented in Appendix A. Temperature generally ranged from 15 to 25–30 °C, following a seasonal pattern and with some shallow stations (BB, BC, PE) reaching the highest summer maxima. Average temperature and standard deviation (SD) of SSW was 21.3 ± 4 °C, similar to BSW of 21 ± 4 °C. Dissolved oxygen ranged from 2.5 to 11, and was not simply driven by temperature. On average, DO was significantly less in BSW than SSW: SSW 6.9 ± 1.4 mg/L and BSW 5.9 ± 1.4 mg/L (Wilcoxon rank sum test, *p* = 2.7 × 10^−5^). Salinity ranged from 22 to 32 ppt, being highest at MB, BB, and SB and lowest at RB. Average salinity of SSW 27 ± 2.5 ppt and BSW 27.3 ± 2.3 ppt were similar. Total chlorophyll ranged between 2 and 120 µg/L with values above ~30 only detected in some RB, as well as OB and MB BSW samples. Mean total chlorophyl was slightly greater in SSW than BSW: SSW 15.1 ± 19.5 µg/L and BSW 13.3 ± 10.4 µg/L, but not significantly different. As expected due to seasonal influence, temperature and dissolved oxygen were significantly different by month (Kruskal–Wallis rank sum test with BF correction). Salinity of both SSW and BSW, as well as chlorophyll of SSW, were significantly different by site using the same test. Pairwise comparisons using Wilcoxon rank sum test with BH correction revealed significant differences by site for salinity, but not for chlorophyll, with the differences by site for salinity consistent between SSW and BSW. Significant differences by year are discussed below in context with analyses on *M. quahogii* abundance data.

### 3.2. M. quahogii in Hard Clams

*M. quahogii* was detected in hard clam tissue samples at every sampling time point (Figure 3 and Appendix A, *n* = 977). Overall, 30.9% of clams were positive (quantifiable), 43.8% were positive but BLD, and 25.3% were negative for *M. quahogii*. No clams showed gross signs of QPX disease and *M. quahogii* prevalence by histopathology was low with only one clam positive for QPX disease in 2014 at RB21 out of 74 (1.4%) analyzed and in 2015, 6 of 45 clams (13.3%) were positive, which were from RB8, RB16, and MA (Appendix A). By qPCR, *M. quahogii* intensity or weighted prevalence (WP) was also low, with a mean value of 1.15 and range from 0.15 to 2.8, representing a rare to mild parasite load (Appendix A). In fact, the mean WP resided between rare and light, with most samples below the limit of detection of the qPCR assay. *M. quahogii* WP followed a similar seasonal trend as prevalence. Mean *M. quahogii* prevalence and weighted prevalence were similar between New York (NY) and Massachusetts (MA). Mean QPX prevalence for NY and MA was 75% and 69%, respectively, and mean *M. quahogii* weighted prevalence for NY and MA was 1.15 and 1.24, respectively. Additional analyses on these samples are discussed below in relation to *M. quahogii* in the environment.

In hard clam pallial fluid (*n* = 291), *M. quahogii* prevalence, the sum of positive and BLD samples, ranged from 28 to 100% with an average of 73% (Appendix A). Mean WP was 1.21 with a range of 0.28 to 2, suggesting rare to light intensity (concentration) of *M. quahogii* in pallial fluid. Mean *M. quahogii* concentration (of positive samples only) was 1587 copies/mL ± 1889 (SD) with a range of 502 to 53,080 copies/mL pallial fluid. Most pallial fluid samples contained pseudofeces (*n* = 280, 96%) and some samples contained sediment (*n* = 165, 57%). There was no significant difference in *M. quahogii* concentration (copies/mL) between samples with and without sediment or pseudofeces (permutation test for independent samples, *p* > 0.05). *M. quahogii* prevalence in hard clam pallial fluid and tissue samples was similar, with 73% of pallial fluid samples and 78% of tissue samples positive (Appendix A). While *M. quahogii* prevalence in pallial fluid and tissue appear similar at the cohort level, at the individual clam level, *M. quahogii* in pallial fluid and tissue had an inverse relationship (Figure 4): when *M. quahogii* was high in the tissue, it was absent or low in the pallial fluid, and vice versa. This ‘high-low’ relationship between pallial fluid and tissue was observed for all sampling sites and months. 

### 3.3. Labyrinthulomycete Composition of Hard Clam Pallial Fluid

The pallial fluid of 23 samples was successfully amplified in the labyrinthulomycete-specific PCR assay and the products were sequenced. Of the 23 products, 14 had a single PCR product (clean chromatograms with only one peak), while nine products were mixed and deconvoluted using BCV, which resulted in 11 sequence ‘clusters’. Sequenced products or BCV clusters were identified using Blastn and phylogenetic analysis was used to validate the labyrinthulomycete group assignment (Figure 5), which agreed between the two methods, despite poor bootstrap confidence values for the phylogenetic tree. *M. quahogii* was found in 74% (*n* = 17) of the pallial fluid samples and 70% (*n* = 16) of the samples contained only *M. quahogii*. The next most prevalent labyrinthulomycete group was oblongichytrids, which were in 26% of samples (*n* = 6) and one sample contained a mix of sequences, representing an aplanochytrid and thraustochytrid. Most samples contained pseudofeces (91%, *n* = 21) and the presence of pseudofeces did not relate to pallial fluid labyrinthulomycete composition or the presence of *M. quahogii*.

### 3.4. M. quahogii and Labyrinthulomycetes in the Environment

*M. quahogii* was detected at every sampling time point in at least one of the environmental sample types, except at BC on 14 July 2015 (SED and SSW were negative and there was not a BSW sample due to the tides). Overall, 75% of environmental samples and 75% of clam samples were positive for *M. quahogii*, showing similar prevalence in host and the environment. *M. quahogii* was less prevalent (Figure 2) and abundant in SSW than SED and BSW (Figure 6 and Appendix A), although differences in prevalence were not statistically significant. There was a change between years in which sample type had greatest prevalence, from 100% prevalence in SED in 2014 to 100% prevalence in BSW in 2015 (Figure 2). To examine whether this might be an artifact of sample collection, we tested whether the detection of *M. quahogii* in BSW was associated with the presence of sediment in BSW samples, but found no relationship (although there were proportionally more samples with sediment in BSW in 2015 (48%) compared to 2014 (20%)), suggesting it may be a true biological signal of interannual variability. Labyrinthulomycetes were ubiquitous, detected in all samples (100% prevalence; Appendix A). *M. quahogii* comprised a small fraction of the labyrinthulomycete community (Appendix A) with an overall mean of 0.55%. The percentage of *M. quahogii* (QPX) was greatest in SED and BSW, reaching as high as 5.68% in SED and 12.23% in BSW.

### 3.5. Conversion to Cellular Concentration

Gene copies can be converted to a theoretical mononuclear cell count by applying a conversion of 440 copies/cell for *M. quahogii* (Table 2), as determined by [17]. *M. quahogii* had a mean concentration of 8 cells/mg tissue in hard clam tissue, 3.6 cells/mL in hard clam pallial fluid, 0.211 cells/mg SED, 0.548 cells/mL BSW, and 0.0168 cells/mL SSW. *M. quahogii* reached as high as 1502 cells/mg tissue in hard clam tissue and 6.77 cells/mL in BSW. 

### 3.6. Group Comparisons

There were no significant differences for *M. quahogii* prevalence in hard clams (both tissue and pallial fluid) or abundance in the environment by QPX disease history or sampling site when Bonferroni (BF) correction was applied (Appendix A). *M. quahogii* prevalence (%TPOS = total positive, includes positive and BLD samples) in hard clam pallial fluid was significantly different by QPX disease history if BF was not applied (*p* = 0.0229) and interestingly, *M. quahogii* prevalence in clams was higher at sites without a previous history of QPX disease (Appendix A). 

In hard clams, *M. quahogii* prevalence was not different by sampling month for either tissue or pallial fluid samples with BF correction (Appendix A). As an exploratory analysis, if we did not apply the *p*-value adjustment, *M. quahogii* %POS (*p* = 0.0356) and WP (*p* = 0.0411) in tissue were different by month. Pairwise comparisons using Wilcoxon rank sum test without *p*-value adjustment showed that June had greater values than September and October for both clam tissue parameters. Similarly, in pallial fluid, *M. quahogii* %TPOS (*p* = 0.0446) and %BLD (*p* = 0.0444) prevalence were also significantly different by month without BF correction; for both pallial fluid prevalence parameters, *M. quahogii* in June was significantly higher than May without *p*-value adjustment. In environmental samples, *M. quahogii* concentration was not significantly different by sampling month with or without *p*-value adjustment for all sample types (Appendix A). 

By year, *M. quahogii* abundance was significantly different in sediment (*p* = 6.25 × 10^−11^) and BSW (*p* = 6.24 × 10^−5^), and not different in hard clams or SSW (Appendix A). There was more *M. quahogii* in sediment in 2014 and in BSW in 2015. Additionally, seawater sample types (BSW vs. SSW) were different only in 2015 (*p* = 3.2 × 10^−9^), with more *M. quahogii* in BSW in 2015. 

Since *M. quahogii* in the environment was so markedly different between the two years in sediment and BSW, we examined all of our metadata for differences between years. Of our 47 quantitative metadata parameters, 11 were significantly different between the two sampling years (Appendix A), which included salinity, chlorophyll, wind direction, and precipitation. The only significant parameters after BF correction for multiple comparisons were precipitation 3 and 4 month means and sums, as well as the difference between salinity in SSW and BSW. There was more precipitation in 2014 (36.22 ± 11.79 cm) compared to 2015 (25.07 ± 7.24 cm), which is reflected in the salinity data for both surface and bottom seawater (salinity is lower in 2014 and higher in 2015 by 1.57 for SSW and 1.05 ppt for BSW). Additionally, in 2014 the difference in salinity between surface and bottom seawater (SSW-BSW) had a greater range compared to 2015, with a mean value of −0.5 ppt in 2014 and 0.02 ppt in 2015. 

### 3.7. Correlations

There were only weak correlations (|rho| < 0.3) between *M. quahogii* in hard clam tissue and the environment in analyses with both years (Figure 7). *M. quahogii* in BSW was positively correlated with *M. quahogii* %POS in hard clam tissue (rho = 0.25, *p* < 0.1) and *M. quahogii* in SSW was significantly negatively correlated with *M. quahogii* %BLD in hard clam tissue (rho = −0.27, *p* < 0.05). In hard clam pallial fluid, *M. quahogii* WP was significantly negatively correlated with *M. quahogii* in BSW (*p* = −0.561, *p* = 0.016; not shown). There were also weak correlations between *M. quahogii* and labyrinthulomycete abundance in environmental samples (Appendix A). *M. quahogii* and labyrinthulomycete abundance in sediment (rho = 0.30, *p* < 0.05) and BSW (rho = 0.37, *p* < 0.01) were significantly positively correlated, but not in SSW.

Correlation analyses were also performed to compare metadata with *M. quahogii* in hard clams and the environment (Appendix A). Only correlations with *p* < 0.05 were considered with or without BF *p*-value correction; if multiple parameters of the same index (e.g., temperature monthly means or lags) were correlated, we only considered the parameter with the strongest correlation. Although *M. quahogii* in clams was not significantly different by month with BF correction (see above), clam *M. quahogii* prevalence and concentration parameters in tissue were moderately negatively correlated with month, day of year, and a temperature parameter (rho ranged from −0.317 to −0.428), with the correlations with temperature significant with BF correction. *M. quahogii* in BSW also had a negative correlation with seasonal parameters (day of year and month), but was positively correlated with a temperature parameter, although these correlations were not significant with *p*-value adjustment. Mean clam tissue *M. quahogii* concentration of the clam cohorts was also positively related to BSW DO (rho = 0.359, *p* = 0.005) and negatively related to BSW chlorophyll b concentration (rho = −0.275, *p* = 0.03); minimum clam tissue *M. quahogii* concentration of the clam cohorts was negatively related to mean wind speed (rho = −0.275, *p* = 0.04). *M. quahogii* %BLD in clam tissue had different relationships from the other *M. quahogii* clam parameters in tissue: it was not correlated with day or month and negatively related to precipitation (rho = −0.487, *p* = 0.00009). There were very few correlations of metadata with *M. quahogii* prevalence and abundance in hard clam pallial fluid and none were significant with BF correction. Without *p*-value adjustment, *M. quahogii* in pallial fluid was positively correlated to salinity and negatively to precipitation and mean wind speed. *M. quahogii* abundance in SSW was not correlated to any environmental parameter. In BSW, *M. quahogii* abundance was correlated to at least one environmental parameter in each category except salinity and precipitation, which were correlated with *M. quahogii* abundance in sediment (negative and positive correlations, respectively). *M. quahogii* in sediment had a negative correlation with BSW chlorophyll a concentration (rho = −0.28, *p* = 0.019), while *M. quahogii* in BSW was significantly, positively correlated (rho = 0.444, *p* = 0.0001). 

## 4. Discussion

This is the first field survey to quantitatively examine *M. quahogii* in clams and the environment, and it revealed that *M. quahogii* was prevalent in hard clams and the environment (Figure 2) both at sites with and without a known history of QPX disease (Figure 3 and Figure 6). *M. quahogii* was almost always detected in BSW and SED (Figure 6), suggesting the flocculent layer at the sediment–water interface may be *M. quahogii’s* preferred habitat, making interaction with hard clams likely as they live in the same habitat buried in the sediment. Only weak correlations were found between *M. quahogii* in hard clams and the environment (Figure 7), which was supported by multivariate analyses described in [16]. These results are not characteristic of an obligate pathogen, for which we would expect a direct relationship between *M. quahogii* in clams and the environment. For example, the abundance of *Perkinsus marinus* (an obligate pathogen of the eastern oyster) in the water column is significantly positively correlated with *P. marinus* weighted prevalence in oysters, as well as oyster mortality [32]. Furthermore, no differences were detected between sites with and without a known history of QPX disease (Appendix A and Appendix A), similar to a non-quantitative study that also found no difference in *M. quahogii* prevalence in a variety of sample types at locations with and without active QPX disease [13]. This suggests that *M. quahogii* in the environment is not a determining factor of QPX disease, and that other factors influence disease dynamics, which is more characteristic of an opportunistic pathogen.

*M. quahogii* was found in 75% of environmental samples and 75% of clam tissue samples (Figure 2), but at low abundance (maximal abundances of *M. quahogii* were estimated as 0.4 cells/mL SSW, 6.77 cells/mL BSW, and 1.6 cells/mg sediment; Table 2). In addition, *M. quahogii* usually represented less than 1% of total labyrinthulomycetes (Appendix A and Appendix A), which were ubiquitous in these environmental samples (Appendix A and Appendix A). Using the same conversion to cellular abundance applied to the *M. quahogii* abundance data (440 copies/mononucleate cell), labyrinthulomycete abundance in sediment ranged from 1.11 to 158 cells/mg, BSW ranged from 1.64 to 391 cells/mL, and SSW ranged from 1.78 to 437 cells/mL. Low abundance (<5 *M. quahogii* cells per slide) and patchy distributions were found by [13] using *in situ* hybridization of environmental samples from Massachusetts, supporting our abundance estimates by nqPCR. The percentage of *M. quahogii* was significantly different by sample type [16], where *M. quahogii* comprised a greater proportion of the labyrinthulomycete community in sediment and BSW compared to SSW, reaching as high as 5.68% in SED and 12.23% in BSW (Appendix A).

### 4.1. Site-Specific Differences 

During this study, *M. quahogii* had high-prevalence but low-intensity in clam tissue. By qPCR, there were no apparent differences for *M. quahogii* prevalence in clams by site (Appendix A). By histopathology, only clams from RB and MA were positive, suggesting that: (1) there is no active QPX disease at the other sites, or (2) QPX infections were extremely light or focal and therefore missed by histopathology. *M. quahogii* abundance in the environment was also not significantly different by site (Appendix A), while labyrinthulomycetes were different by site in SED and SSW [16]. Since salinity was significantly different by site [16], it may explain the observed differences in labyrinthulomycete abundance (Appendix A), as zoosporulation is typically repressed at salinities above 15 ppt for culturable thraustochytrids [33,34], consistent with negative correlations between abundance and salinity in other studies [35,36]. This is consistent with this study, as labyrinthulomycete abundance in sediment was greater at OB, which was also less saline, compared to MB (Appendix A). Similarly for SSW, PE had a lower abundance and was more saline compared to BB, MB, and RB, and RB had a greater abundance and was less saline compared to SB. In contrast, *M. quahogii* did not exhibit these relationships, despite known effects of salinity on *M. quahogii* in vitro. In culture, growth by endosporulation [37,38] and zoosporulation [1] has an inverse relationship with salinity with suppression observed at lower salinities; however, differences in salinity among sites did not seem to be related to *M. quahogii* abundance in this study.

### 4.2. Seasonal Variability

For *M. quahogii* in clams, there were seasonal differences or influences suggested by multiple analyses (Appendix A), including multivariate analyses in [16]. Although subtle, there is a general trend in the data of higher *M. quahogii* prevalence and WP in clams during late spring and early summer (months) dissipating through the fall (months) (Figure 3), consistent with seasonal findings from other studies in New York on QPX disease [39,40]. Seasonal influence is also suggested by the correlations with metadata that are seasonally influenced, such as temperature, dissolved oxygen, chlorophyll, and wind (Appendix A). Temperature had an inverse relationship with both *M. quahogii* prevalence and concentration in clam tissue (Appendix A). In our analyses, the 3-month mean or lag mean air temperature (e.g., mean temperature from the 60 to 90 day interval before the sampling date) was more strongly correlated with *M. quahogii* prevalence or concentration in clam tissue than any other temperature metric (Appendix A). Although the correlations were not strong, they are supported by [39], which found a strong correlation between *M. quahogii* weighted prevalence in clams and seawater temperature 120 days before the clam sampling date (R^2^ = 0.986) at a site in Raritan Bay. The temperature of previous months may coincide with the time of infection and is consistent with the slow and temperature-dependent progression of QPX disease [41,42]. This suggests that the temperature of previous months could potentially be used as an indicator to forecast or predict QPX disease, which was also suggested by [43], although this interpretation is based on data collected during the normal field season (e.g., April/May to October/November) and may not be necessarily extrapolated to winter trends.

In general, the greatest abundance for both *M. quahogii* and labyrinthulomycetes in the environment occurred early in the field season, from late spring to summer, with usually lower abundance found in fall (Figure 6 and Appendix A), although for *M. quahogii* differences by month were not significant (Appendix A). There may be a seasonal signal, particularly for *M. quahogii* in BSW, as there were weak negative correlations with month and day (Appendix A); however, this may have been driven by interannual variation (discussed below). This subtle “seasonal” trend exhibited by *M. quahogii* in the environment was also the general trend for *M. quahogii* in hard clam tissue (Figure 3), suggesting that *M. quahogii* in clams and the environment follow similar seasonal trajectories.

For labyrinthulomycetes in seawater, the negative correlations with month and day were stronger compared to *M. quahogii* [16]. Further supporting an element of seasonality are the correlations of *M. quahogii* (and labyrinthulomycete) abundance with seasonally influenced environmental parameters such as temperature, dissolved oxygen, chlorophyll, and wind (Appendix A). This “seasonal” trend is similar to that observed by other studies on labyrinthulomycetes with peaks in abundance occurring from spring to late summer followed by lows in fall and winter [35,44,45,46]. In sediment, the lack of seasonal influence for both *M. quahogii* and labyrinthulomycetes is consistent with other studies, which suggests that abundance in the sediment is more stable and exhibits less seasonality compared to seawater [4,47,48,49]. These results, combined with the significant positive correlations between *M. quahogii* and labyrinthulomycete abundance in BSW and sediment (Appendix A), suggest that *M. quahogii* behaves similarly to the rest of the labyrinthulomycete community. 

### 4.3. Interannual Variability

The two years of this study captured some distinct conditions, including differences in QPX disease prevalence by histopathology, with only 1.4% positive in 2014 and 13.3% positive in 2015 (Appendix A). These interannual differences were also evident in multivariate analyses described in [16] without BF correction for multiple comparisons. The mean, median, and range of all *M. quahogii* clam prevalence parameters (%TPOS, POS, BLD, and WP) were greater in 2015 compared to 2014, shown in [16]. Salinity and precipitation parameters were significantly different between the two years (Appendix A and Appendix A), suggesting that 2014 was a “wetter” year (more freshwater, lower salinity) compared to 2015 (less freshwater, higher salinity). 

For *M. quahogii* in the environment, there was significant interannual variability in sediment and BSW. As suggested with *M. quahogii* in clams, this interannual variability may be driven by differences in precipitation that can affect many other environmental factors, including salinity, delivery of terrestrial organic matter, and production of marine organic matter. As with *M. quahogii* in sediment, labyrinthulomycetes also exhibited a higher mean, median, and range in sediment in 2014, which decreased in 2015 coinciding with a slight increase in abundance in BSW (Appendix A) [16]. This suggests similar trends or responses to environmental factors between *M. quahogii* and labyrinthulomycetes. This may reflect the input of terrestrial runoff via precipitation given the estuarine and coastal nature of the sampling sites, to bringing allochthonous organic matter into the marine system, which *M. quahogii* and/or other labyrinthulomycetes could take advantage of. 

### 4.4. Chlorophyll

*M. quahogii* abundance in BSW was positively correlated with chlorophyll in BSW (Appendix A). This relationship was also significant between total labyrinthulomycetes in both SSW and BSW, with the relationship stronger in BSW compared to SSW [16]. Other studies have also found positive relationships between labyrinthulomycetes and chlorophyll [36,45,50,51], suggesting that labyrinthulomycetes, including *M. quahogii*, at our sites may thrive on autochthonous (marine) organic matter, particularly in BSW. Further support for this interpretation comes from in vitro investigations that have shown that *M. quahogii* can grow on macroalgae [13,52], which may be an environmental reservoir or substrate that *M. quahogii* can exploit in the absence of the hard clam host.

### 4.5. Hard Clam Pallial Fluid and Tissue

To our knowledge, this is the first study to examine hard clam pallial fluid for *M. quahogii*. Similar to *M. quahogii* in hard clam tissue, there were no significant differences by QPX disease history or sampling site for *M. quahogii* in pallial fluid, with some weak signs of seasonality (Appendix A). Although prevalence of *M. quahogii* in hard clam tissue and pallial fluid were similar at the cohort level, 78 and 73%, respectively (Appendix A), there was a clear and surprising inverse, ‘high-low’ relationship between *M. quahogii* in pallial fluid and tissue at the individual clam level (Figure 4). Furthermore, *M. quahogii* WP in pallial fluid was negatively correlated with *M. quahogii* in BSW. If *M. quahogii* was a transient component of pallial fluid, then we would expect to see a direct relationship rather than an inverse relationship. The average concentration of *M. quahogii* in BSW (0.04 cells/mL ± 0.07 SD) does not account for the average concentration observed in pallial fluid (3.6 cells/mL ± 4.3 SD), suggesting that either hard clams concentrate *M. quahogii* in the pallial fluid or that *M. quahogii* actively stays and grows within the pallial cavity. In the pallial cavity, *M. quahogii* may colonize mucosal surfaces as a habitat. It may also use pseudofeces, which are formed and transit throughout the pallial cavity, as a substrate for growth [10,53]. The specificity of this relationship is supported by the fact that *M. quahogii* is a minor component of the total labyrinthulomycete community outside of the clam (Appendix A and Appendix A), but the dominant labyrinthulomycete in hard clam pallial fluid (Figure 5)The lack of similarly correlated environmental parameters for *M. quahogii* in hard clam pallial fluid and tissue (Appendix A) suggests that *M. quahogii* is influenced by different environmental factors depending on host anatomical location. 

Phylogenetic analyses of labyrinthulomycete PCR products from the field hard clam pallial fluid samples revealed that *M. quahogii* was the dominant labyrinthulomycete with 70% of samples containing only *M. quahogii* (Figure 5). Hard clam pallial fluid field samples also harbored oblongichytrids and one sample contained a mix of aplanochytrids and thraustochytrids. Although pathogen reservoirs are usually considered to be secondary hosts or substrates in the environment, this study suggests that pallial fluid may represent a *M. quahogii* ‘reservoir’ within its primary host. In this light, *M. quahogii* could exist and concentrate in pallial fluid, perhaps as a commensal, until it can opportunistically breach host barriers and penetrate and infect the clam tissue, as a function of host immune status or changes in the host–microbe–environment relationship. In fact, many labyrinthulomycetes, particularly thraustochytrids, tend to form host- or substrate-specific associations with marine animals that are thought to be mutualistic or commensal in nature [4,47,54], with the potential to become opportunistic pathogens [2,3]. Alternatively, the inverse, ‘high-low’ relationship found for *M. quahogii* in hard clam tissue and pallial fluid may represent an effective immune response by the clam, which enhances the production of mucosal immune effectors [55] and purges *M. quahogii* from the pallial cavity once *M. quahogii* breaches host barriers and invades the tissue.

## 5. Conclusions

*M. quahogii* is broadly distributed in hard clams and the environment, in both areas with and without a known history of QPX disease. Active QPX disease was only confirmed by histopathology at Raritan Bay and Massachusetts, which are considered QPX-enzootic sites with an intense history of QPX disease and hard clam mortality events. *M. quahogii* prevalence and intensity in clams was only weakly related to *M. quahogii* in the environment, supporting that *M. quahogii* is a commonly distributed opportunistic pathogen, where pathogenesis is initiated when clams are disadvantaged by “unfavorable genotype-environment interactions” [14], leading to host immune suppression and QPX disease [2,3,8,9]. The consistent detection of *M. quahogii* in sediment and bottom seawater suggests that the flocculent layer at the sediment–water interface may be *M. quahogii’s* preferred habitat, making interaction with hard clams likely as they live in the same habitat buried in the sediment. As illustrated by the ‘high-low’ relationship between *M. quahogii* in hard clam pallial fluid and tissue, *M. quahogii* was always present in its hard clam host in either the tissue or pallial fluid, although different environmental factors and/or immune abilities may influence the prevalence and/or intensity of *M. quahogii* in different host anatomical locations. Furthermore, *M. quahogii* was the most common labyrinthulomycete in hard clam pallial fluid samples despite being a minor component of the external labyrinthulomycete community, supporting a unique or host-specific relationship between *M. quahogii* and the hard clam. The novel results obtained from hard clam pallial fluid further support the hypothesis that *M. quahogii* is a commensal or avirulent member of the hard clam microbiota until perturbations in host immunity or the host–microbe–environment relationship result in virulence and pathogenesis [2,3,56], supporting its classification as a commensal, opportunistic pathogen.

This study also suggests there is minimal risk of spreading *M. quahogii* to receiving bays through transplant or restoration programs because *M. quahogii* is already present (at least in all of our NY field sites) and does not appear to be causing disease in hard clam populations in these locations. There was slight seasonal variation, and some signs of interannual variation of *M. quahogii* prevalence and intensity by qPCR and histopathology in hard clams, compared to weak signs of seasonal variation and marked interannual variability of *M. quahogii* in the environment (sediment and bottom seawater), which may be attributed to variations in precipitation (wet versus dry years). *M. quahogii* in clams and the environment showed similar general trends in abundance, with highest prevalence or abundance in early spring to summer and least in late summer to fall, which was also similar for labyrinthulomycetes in the environment. While *M. quahogii* was prevalent in the environment, abundance was low compared to total labyrinthulomycetes and usually comprised less than 1% of the community. *M. quahogii* appeared to behave similar to labyrinthulomycetes outside of the hard clam host, although overall showed less seasonality compared to the whole community, which was more dynamic in seawater than sediment. Further understanding of QPX disease in hard clams requires more long-term monitoring and surveillance, particularly at QPX-enzootic sites, with the addition of environmental conditions prior to the sampling date with means, sums, or lags (at least up to 120 days prior), in order to better decipher QPX disease dynamics.

## Figures and Tables

**Figure 1 jof-08-01128-f001:**
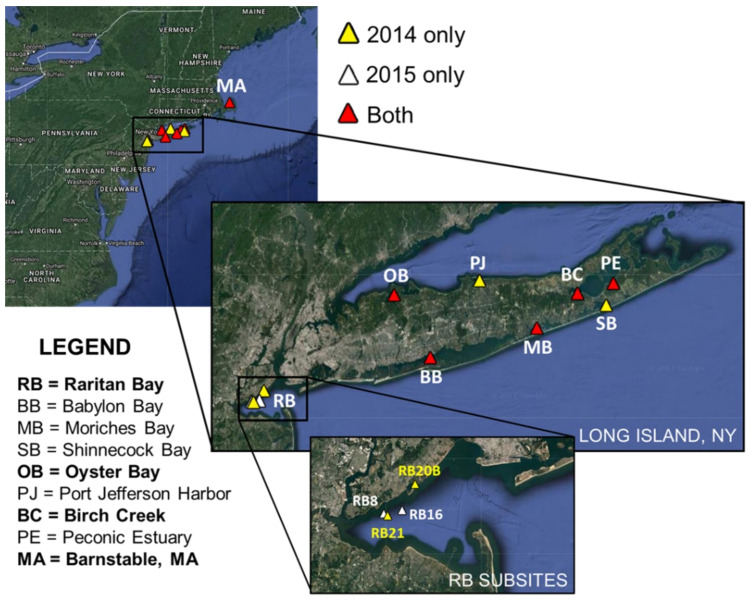
Location of sampling sites. Between 2014 and 2015, 12 sites were surveyed throughout the marine district of New York, including a control site in Barnstable, Massachusetts (MA). Four different locations within Raritan Bay (RB) were surveyed between the 2 years. Sites in bold on the legend (RB, OB, BC, and MA) have a previous history of QPX disease.

**Figure 2 jof-08-01128-f002:**
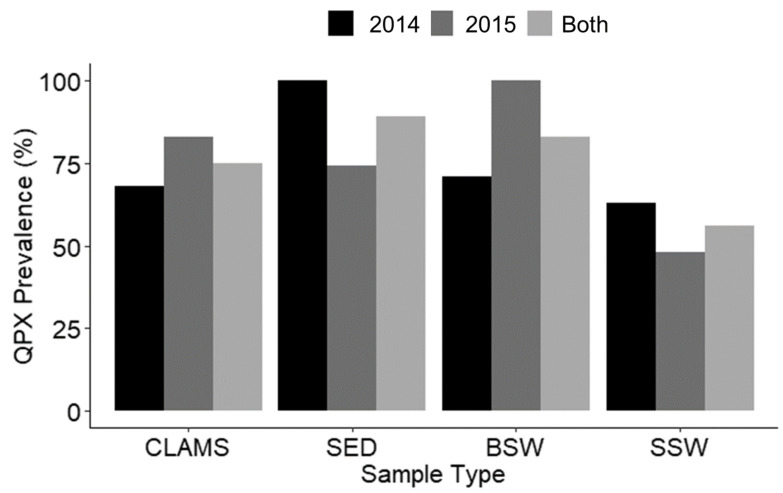
Summary of *M. quahogii* prevalence (%QPX-positive samples) by sample type and year. CLAMS: tissue samples, SED: sediment, BSW: bottom seawater, SSW: surface seawater.

**Figure 3 jof-08-01128-f003:**
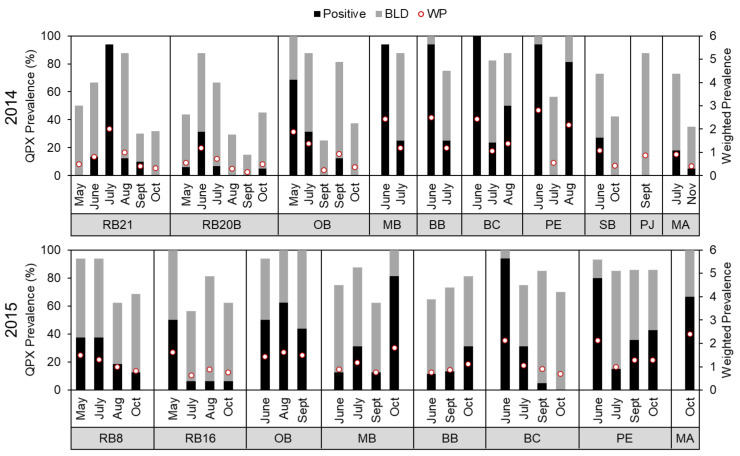
*M. quahogii* (QPX) prevalence and weighted prevalence (WP) in hard clam tissue samples as determined by qPCR by sampling site and month for 2014 (**top**) and 2015 (**bottom**). Below limit of detection (BLD) represents samples that were positive but could not be accurately quantified. The percentage of positive, quantifiable and positive, BLD samples represents total *M. quahogii* prevalence.

**Figure 4 jof-08-01128-f004:**
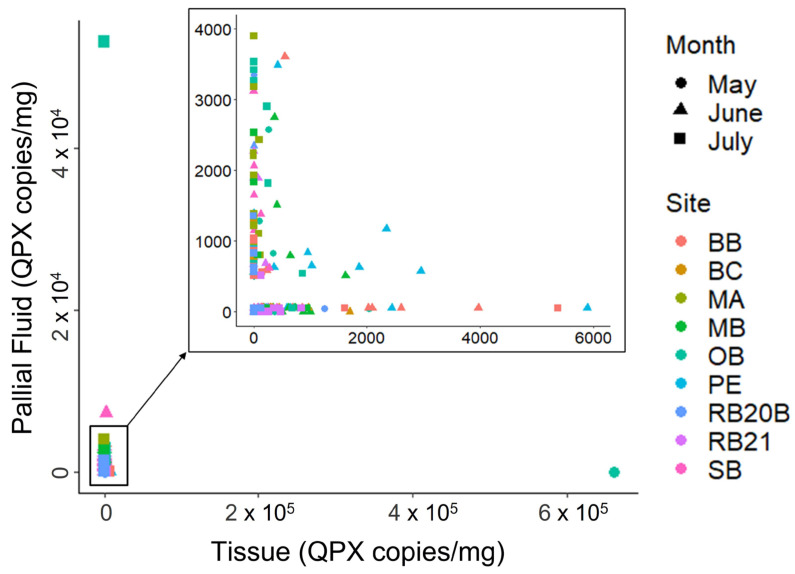
*M. quahogii* (QPX) in hard clam pallial fluid and tissue for each individual clam. Negative samples were assigned a value of 0 and BLD samples were assigned 10% of the qPCR assay’s limit of detection (LOD) for the specific sample type (50 copies/mL for pallial fluid BLD and 7.5 copies/mg for tissue BLD) for visualization.

**Figure 5 jof-08-01128-f005:**
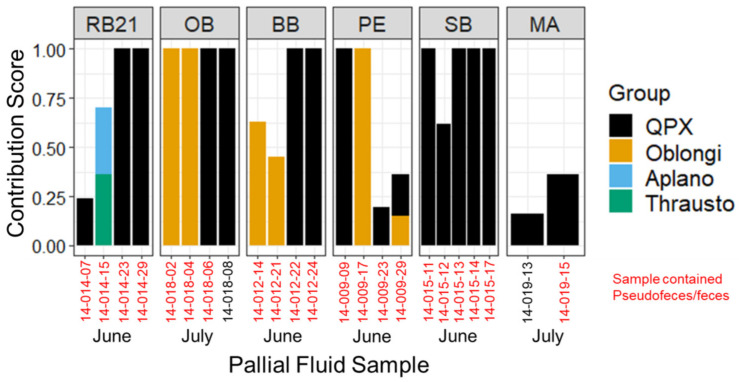
Labyrinthulomycete composition of hard clam pallial fluid. The *y*-axis represents the expected BCV contribution score for mixed sequences that were deconvoluted, while sequences containing a single dominant product were assigned the value of 1. QPX = *M. quahogii*; Oblongi = oblongichytrid; Aplano = aplanochytrid; Thrausto = thraustochytrid.

**Figure 6 jof-08-01128-f006:**
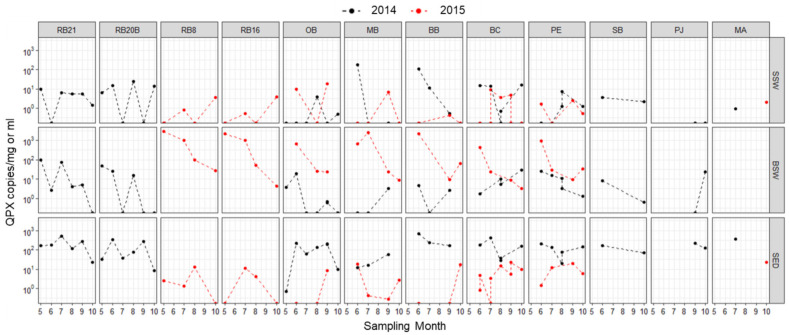
*M. quahogii* (QPX) abundance in environmental samples: surface seawater (SSW), bottom seawater (BSW), and sediment (SED), assayed using nqPCR. Values are expressed in terms of QPX gene copies per mL seawater or mg sediment on a log10 scale.

**Figure 7 jof-08-01128-f007:**
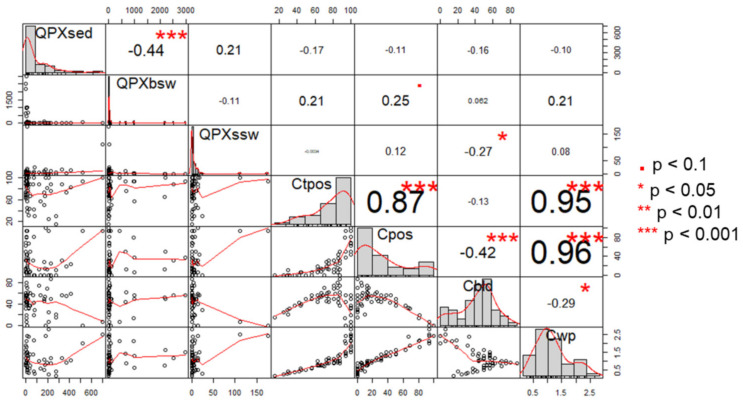
Spearman correlation coefficients (rho) correlogram with histograms for *M. quahogii* (QPX) in hard clam tissue (Ctpos, Cpos, Cbld, and Cwp represent clam tissue QPX prevalence: %TPOS, %POS, and %BLD and WP), and *M. quahogii* in the environment (QPXsed, QPXbsw, and QPXssw represent QPX abundance in sediment, BSW, and SSW).

**Table 1 jof-08-01128-t001:** Summary of samples collected during the two-year field survey.

Sample Type	2014	2015	Total
Hard Clams	539	438	977
Surface Seawater (SSW)	40	31	71
Bottom Seawater (BSW)	38	26	64
Sediment (SED)	40	31	71
TOTAL	657	526	1183

**Table 2 jof-08-01128-t002:** Mean and range of *M. quahogii* (QPX) concentration converted to theoretical cellular concentration (cells/mg tissue, sediment or mL pallial fluid, seawater) from gene copy number based on a conversion factor of 440 copies/mononucleated *M. quahogii* cell. CLAM TIS, clam tissue samples; CLAM PF, clam pallial fluid samples.

Statistic	Sample Type	2014	2015	Both Years
Mean ± Standard Deviation	CLAM TIS *	12 ± 126	3 ± 20	8 ± 90
	CLAM PF *	3.6 ± 4.3	n/a	n/a
	SED	0.364 ± 0.34	0.0148 ± 0.02	0.211 ± 0.31
	BSW	0.267 ± 0.05	1.311 ± 2	0.548 ± 1.43
	SSW	0.258 ± 0.07	0.005 ± 0.01	0.0168 ± 0.06
Range (Minimum, Non-zero Minimum—Maximum)	CLAM TIS	0, 0.2–1502	0, 0.2–191	0, 0.2–1502
	CLAM PF	0, 0.04–121	n/a	n/a
	SED	0.2–1.6	0, 0.0006–0.05	0, 0.0006–1.6
	BSW	0, 0.0013–0.23	0.007–6.77	0, 0.0013–6.77
	SSW	0, 0.0011–0.4	0, 0.001–0.04	0, 0.001–0.4

* Mean and standard deviation of clam samples is of the positive, quantifiable samples only and excludes negative and BLD samples.

## Data Availability

Not applicable.

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
