# Peer review of "Mucochytrium quahogii (=QPX) Is a Commensal, Opportunistic Pathogen of the Hard Clam (Mercenaria mercenaria): Evidence and Implications for QPX Disease Management"

_jof, 2022, doi:10.3390/jof8111128_

Round 1

Reviewer 2 Report

This is a very valuable contribution clearly demonstrating that Mucochytrium quahogii is commonly associated with hard clams and their estuarine environment, even in the absence of apparent QPX disease. The approach combining environmental eDNA sampling, qPCR detection in clams (both extrapallial fluid and mantle/siphon tissues), and confirmatory histology is very strong. There are important management implications here, regulators tasked with intra- and interstate transport of clam seed should worry less about introducing M. quahogii to areas receiving clam seed as this pathogen is likely endemic in most areas. What triggers M. quahogii pathogenicity remains a mystery but this current paper and much work from your research group has well demonstrated interactions between host and environment as the culprit. I fully recommend this manuscript for publication, I do have two quick minor thoughts that may help readers interpret your results. First, it may be better to plot Fig 4 on a log-log scale, it may help highlight the negative association between QPX copies in tissue and extrapallial fluid, it may even linearize this relationship. Either way this is a really interesting result and highlights potential pathways toward pathogenicity. The other is to simply refer to M. quahogii as a facultative rather than opportunistic pathogen. The latter implies immunosuppression on the side of the host, which may be the case but not without considering the role of the environment. So better to say facultative. Otherwise, congratulations and thanks for this contribution, I will cite this paper often.      

Round 2

Reviewer 1 Report

Journal of Fungi

Manuscript Number: JoF-1917028

Title: Mucochytrium quahogii (=QPX) is a commensal, opportunistic pathogen of the hard clam (Mercenaria mercenaria): evidence and implications for QPX disease management.

Reviewer

General Comment:

The manuscript deals with a survey done in 2014-2015 in an area in New York and a control area in Massachussetts, where hard clam inhabits and presence of a commensal, opportunistic pathogen (Mucochytrium quahogii) is also present. The aim of the study was to determine whether a relationship exists between presence of QPX in water, sediments and clams, in the study area, and to quantify the load of QPX in clam tissues, pallial fluid and in environmental samples. As QPX is a commensal, opportunistic pathogen of hard clams, no direct relationship was observed between presence in tissues and environmental samples. In fact, QPX was present in all sites sampled, but not in all those sites, the QPX disease appeared. This confirmed the notion that such species is an opportunistic pathogen, and the onset of disease may be related to environmental and clam physiological conditions. The study also used the pallial fluid and tissues to determine the presence of QPX through molecular methods such as qPCR, nested qPCR, in situ hybridization, and histopathology.

The prevalence and intensity of QPX in clams was weakly related to the environment, supporting that QPX is a commonly distributed opportunistic pathogen. Pathogenesis is initiated when clams undergo unfavorable genotype-environment interactions, leading to host immune suppression and QPX. Authors conclude that detection of QPX in sediment and bottom seawater suggests that the flocculent layer at the sediment-water interface may be the preferred habitat for QPX since this environment is suitable for the interaction with hard clams as they live in the same habitat. QPX was always present in its hard clam hosts in either the tissue or pallial fluid, but varying environmental factors and/or immune status may influence the prevalence and/or intensity of QPX infection in different host anatomical locations.

The manuscript is interesting and it shows novel data on the presence of QPX in different anatomical compartments of the clam. The revised form addressed the previous comments done. Nonetheless still one question arises from this revised manuscript.

Specific comments

Material and Methods

Comment 1 lines 204 - 207. According to the described definition of below detection limit (BDL) positive samples in lines 185-190, the sentence in lines 204-207 indicates that all quantitative data of QPX DNA both by qPCR and nqPCR were expressed as gene copies per mg tissue and/or sediment, but BLD samples could not be quantified. Then, the statement in lines 204-207 should be explained.

Author Response

Clam qPCR data is handled by each sampling cohort explained in lines 180-185 as prevalence (% positive, % BLD) and weighted prevalence, according to the intensity scales (Table S2), see Figure 3.

BLD samples are not considered to be quantitative as per the definition (positive or negative only). Therefore lines 204-207 are referring only to the quantitative data (values) determined by the qPCR or nqPCR assays (e.g., positive, quantifiable samples for the qPCR).

For clarity we have added lines 204-205: “All quantitative data determined by qPCR (i.e., quantifiable results but not BLD) and nqPCR assays is expressed in terms of gene copies…”

In Figure 4, the BLD samples were assigned 10% of the LOD for visualization, which is explained in the caption. In Table 2, it is explained that the statistics for clam samples are based on the positive, quantifiable samples only and excludes negative and BLD samples.